# Use of Online Newborn Screening Educational Resources for the Education of Expectant Parents: An Improvement in Equity

**DOI:** 10.3390/ijns8020034

**Published:** 2022-05-11

**Authors:** Kristen Thompson, Shelby Atkinson, Mary Kleyn

**Affiliations:** Michigan Department of Health and Human Services, Lansing, MI 48933, USA; atkinsons2@michigan.gov (S.A.); kleynm@michigan.gov (M.K.)

**Keywords:** newborn screening (NBS), education, equity, medically underserved areas (MUA)

## Abstract

Educating parents about the newborn screening (NBS) process is critical in ensuring that families are aware of their child’s NBS, which could contribute to better outcomes for the baby and experiences for the family. Successful education efforts result in expecting parents understanding the importance of NBS, feeling comfortable with the NBS process, and being aware of their choices after NBS is complete. Educating parents prenatally is challenging for many NBS programs for a variety of reasons. The COVID-19 pandemic added additional barriers to NBS programs’ ability to educate parents prenatally about NBS. By initiating a department-wide partnership among other programs with a similar target audience, Michigan’s NBS Program was able to host a virtual baby fair. Since the inaugural event, Michigan’s NBS Program has hosted seven virtual fairs with 15 participating programs. A total of 692 participants registered for the baby fair and received a resource packet, over 157 participants joined one of the live presentations, and 211 have viewed the YouTube videos of recorded fairs. Virtual baby fairs are a cost-effective and convenient approach to education that could be implemented in any NBS program to educate parents prenatally about NBS.

## 1. Introduction

Newborn screening (NBS) is an important public health program. Michigan’s public health code requires all newborns to be screened for rare but serious disorders (MCL 333.5431). The Michigan NBS panel currently includes 58 disorders. Infants with these disorders benefit greatly from early treatment, as they appear healthy at birth but can become ill in a very short time. 

While screening is mandated for all newborns in Michigan, educating parents about the NBS process is critical to ensure that families are aware of their child’s NBS, which could contribute to better outcomes for the baby and experiences for the family. Successful education efforts result in expecting parents understanding the importance of NBS, feeling comfortable with the NBS process, and being aware of their choices after NBS is complete. International studies have found that parents generally have limited awareness of NBS [1,2]. Parental awareness that the blood spot screen was collected could improve the notification process when a baby has an abnormal screen [3] and result in more prompt follow-up care for the needed next steps. Providing information to expectant parents may increase satisfaction with the process [4] or lead to increased support for screening [3,5]. 

Since awareness of NBS improves parental experiences, inadequate education about NBS could compound inequities faced by families of medically underserved areas (MUA). A survey conducted by Expecting Health and RTI International showed that families living within MUAs are less likely to receive NBS education prenatally and are less aware of NBS when compared to families living outside of MUAs [6].

The success of any NBS parental education initiative depends on several factors, including knowing when to start providing education about NBS to parents. Michigan’s NBS Program conducted a phone-based survey of parents whose child was diagnosed through NBS, and several parents commented they would have preferred to receive more information about NBS before their child’s birth [7]. Additionally, the American College of Obstetricians and Gynecologists (ACOG) found that expectant parents prefer to receive NBS information during prenatal care visits [8]. ACOG released the following recommendations regarding incorporating NBS education into obstetric practice: 

“*Obstetrician-gynecologists and other obstetric care providers should make resources about newborn screening available to patients during pregnancy. Information can be disseminated through informational brochures and electronic sources and through review or discussion at some time during prenatal care. Integrating education about newborn screening into prenatal care allows parents to be prepared for having their child undergo screening as well as for receiving newborn screening test results*.” (ACOG Committee Opinion No. 778, 2019) [9].

The prenatal timeframe may be the most optimal time for NBS education because it is a time when parents are most eager to learn how they can help their baby, and they have the time to absorb the information [3,8].

In addition to general NBS education, Michigan educates parents about the Michigan BioTrust for Health program, which oversees the storage and use of blood spots that remain after the NBS process is complete. While all blood spots go into storage after screening, parents can choose whether they want their baby’s residual dried blood spots to be available for use for de-identified medical research benefitting public health. Families are asked to document their consent decision about the research use of blood spots around the time that blood spots are collected for NBS. Like NBS education, a study on parental self-reported experiences of BioTrust education also found that parents preferred to receive BioTrust information prenatally [10]. Parents who reported that they received information about the BioTrust program at their doctor’s office or prenatal clinic were significantly more likely to have a completed BioTrust consent form on file after screening, demonstrating the value of providing parents with educational materials during pregnancy (OR 1.4, 95% CI 1.1–1.6) [11]. 

The importance of educating parents prenatally about NBS is a well-established and supported concept among the NBS community [12]. However, Michigan’s NBS team has faced several challenges in connecting with prenatal care providers to increase education for NBS. The Michigan NBS and BioTrust programs have provided support to prenatal providers regarding parent education by hosting vendor booths at conferences attended by prenatal care providers. Since not all providers are able to attend conferences and due to the large geographic size of the state, additional efforts were needed to distribute materials to prenatal care providers across Michigan. Office managers at more than 300 prenatal care offices received a mailing that described the free educational materials the NBS program offers. Despite these efforts to assist clinical partners in educating parents about NBS, the NBS Program cannot monitor the distribution of NBS information. Furthermore, not all pregnant people receive adequate prenatal care, especially when considering medically underserved areas. According to Michigan’s Division for Vital Records and Health Statistics, 32% of pregnant people in Michigan do not receive adequate prenatal care based on the Kessner Index, and 14.7 out of every 1000 pregnant individuals do not receive prenatal care at all [13,14]. 

Because not all pregnant persons receive the full standard of prenatal care, NBS and BioTrust staff need to take a multi-pronged approach to prenatal education and cannot rely on prenatal care providers alone. All hospitals and birthing attendants in Michigan are instructed by the MDHHS to provide parents with NBS and BioTrust education while in their care, including brochures, infographics, and/or a video with information about the programs. Additionally, Michigan maintains program websites with educational resources for parents including information about newborn screening, a timeline of when disorders were added to the Michigan NBS panel, disorder fact sheets, and information about DBS storage and uses through the BioTrust. In addition to the information families should receive in the hospital, NBS and BioTrust staff also attend in-person events. From 2016 to 2020, NBS staff attended 16 baby fairs in 8 of Michigan’s 83 counties to ensure that the most up to date and comprehensive educational materials were being distributed to families (Figure 1a). These events captured only a subset of new parents in Michigan based on the limited number of baby fairs available for attendance and the locations in which they were held. The in-person baby fairs were organized by external organizations which made it hard to accurately track how many attendees received information specifically about NBS and gain insight on the geographic representation of the attendees. Further, fewer families living in MUAs received information about NBS via in-person baby fairs due to a lack of events being held in these areas (Figure 1a) [15].

The COVID-19 pandemic added more barriers to NBS programs’ ability to educate parents prenatally about NBS. As restrictions on in-person gatherings in Michigan were put in place, traditional ways of educating parents, including in-person baby fairs, birth education classes, and hospital tours, were cancelled. The NBS program needed to creatively think of a way to safely educate parents, while following guidelines in place due to the pandemic. In partnership with other areas of the Michigan Department of Health and Human Services (MDHHS), the Michigan NBS Program hosted a virtual baby fair. The virtual baby fair was an opportunity for MDHHS programs, including NBS and the BioTrust for Health, to share important resources and information to new and expecting parents from the comfort and safety of their home. By partnering with other programs, the Michigan NBS Program was able to promote the event to a larger audience and potentially attract more participants who might not otherwise seek out education about NBS. The virtual baby fair allowed the Michigan NBS Program to provide education to new parents across the state from subject matter experts, to ensure optimal consistency and currency in messaging. 

## 2. Materials and Methods

Staff from the Michigan NBS Program reached out to programs within the MDHHS with a similar target audience. From there, a team of representatives from each interested program formed a planning committee. Fifteen programs from the MDHHS participated in the inaugural baby fair, including NBS, BioTrust for Health, Infant Safe Sleep, Women, Infants, & Children (WIC), Immunizations, Early Hearing Detection and Intervention, Birth Defects Education and Outreach, State Breastfeeding Promotion, Maternal Infant Health Program, Childhood Lead Poisoning Prevention Program, Vital Records, Unintentional Injury Prevention, Michigan Home Visiting Initiative, Children’s Special Healthcare Services, and Eat Safe Fish. One event coordinator was identified to lead the efforts of the planning committee. The group met twice a month using a virtual platform, with meetings beginning four months prior to the baby fairs, to discuss logistics and promotion of the event. 

During the first round of virtual baby fairs, two date options were offered for participants: a weeknight and a weekend morning. A “save the date” with registration information was created to provide the public with the information they would need to sign up for the free event. Free promotion of the event was accomplished through MDHHS social media pages, emails to existing listservs, faxes to prenatal care provider offices, and WIC text message notifications. Registration was completed utilizing online survey software. Each registrant was asked to select which fair they planned to attend, provide contact information, and indicate their county of residence.

The planning committee created a packet of information and resources, called the “MDHHS Virtual Baby Fair Resource Packet,” to supplement the information provided at the live events. Each program created a one-page flyer with a summary of the program, what they have to offer parents, educational materials, and contact information. Programs were instructed to include a snapshot of the most important messages they wanted parents to hear about their program to match information that may have been distributed to someone walking up to an informational table at an in-person baby fair. The packet, and any additional program resources, were sent electronically or mailed to registrants, depending on their preference. Everyone who registered for the event received this packet prior to the live event.

An email reminder about the event was sent one week before and the day of each live event. On the day of the MDHHS Virtual Baby Fair, participants joined the event through a video teleconferencing software program, Microsoft Teams. The live baby fair presentation was meant to mimic an interaction someone might have at a traditional in-person baby fair vendor booth. Each program had five minutes to present, followed by two minutes for participants to ask questions. In total, the presentations lasted 1 h and 15 min. Questions were asked in the chat box or emailed if they preferred to ask anonymously. The live event was recorded and made into a YouTube video, a link to this video is listed in the Appendix A. The YouTube link was shared with registrants via email, so if they missed the live presentation or wanted to watch the presentations with closed captioning in their preferred language, they could view it at their convenience later.

After the event, each registrant received an email with the post-event survey and the YouTube link of the recorded event. The post-event survey included questions on how the registrant heard about the baby fair, the ease of using the video teleconferencing software platform, the usefulness of the information presented at the live event and provided in the resource packet, the length of the baby fair, and if other topics should be covered. Feedback on these post-event surveys was used to gain insight on how future events might be improved. 

## 3. Results

In total, 378 people registered for the baby fair and received the Baby Fair Resource Packet. Of those who registered, 64 registrants (17%) requested a hard copy resource packet via mail. Approximately 70 participants joined one of the two live presentations (Appendix A).

Since the first round of fairs, MDHHS has hosted five additional virtual baby fairs. These events were expanded to include additional MDHHS programs and were lengthened to 2 h. The format of the live event, the resource packet, and the promotion of the fairs did not change. To date, 692 new and expecting parents registered for one of the baby fairs and received the Baby Fair Resource Packet. Approximately 15% of all registrants requested a hard copy resource packet. Approximately 157 participants joined one of the live presentations, and 211 have viewed the YouTube videos of recorded fairs.

As part of the resource packet, the NBS program flyer included a description of what NBS is, how many babies are found to have a disorder detected by NBS, a reminder that the parent does not need to request that NBS be performed, what they can expect from the newborn screen, and stressed the importance of identifying a primary care provider for their baby before delivery. The NBS program also included a QR code which linked to a NBS informational video for those who prefer information to be read to them. This video is an animated road map of the newborn screening process, explaining the three parts to the newborn screen, testing of the blood spot at the state laboratory, and that parents should receive the results of their baby’s NBS from their baby’s primary care provider. A flyer was also included for the Michigan BioTrust for Health program to provide families with information about how blood spots are used after screening is conducted and the options they have regarding usage of those spots. The goal of providing this information about the BioTrust to families was to prepare them for the consent decision they will be asked to make shortly after screening takes place at the hospital. Contact information for each program was listed, and links to the NBS and BioTrust websites were provided so participants could access additional resources. 

The NBS and BioTrust programs’ live presentations included an overview of what NBS is, how many babies are diagnosed with a NBS condition in Michigan, an overview of the three components of NBS, with pictures showing a newborn having the hearing screen and pulse oximetry screens conducted and the blood spot screen collected. Additional information included what parents can expect at each step of the NBS process (before delivery, during their hospital stay, and after discharge), parents’ choices regarding the research use of leftover blood spots, and an overview of program brochures and infographics. Those educational materials were distributed to each registrant in the resource packet so that they could reference them after the baby fair.

The virtual baby fairs have attracted registrants from 44 of Michigan’s 83 counties (Figure 1b). Of those 44 counties, 35 counties contain MUAs [15].

A post-event survey was sent to all registrants. Of the 692 registrants, 40 completed the survey (6%). All 40 respondents found the baby fair to be helpful to them as new or expecting parents, with 40% finding it somewhat helpful and 60% finding it very helpful. Among all respondents, 88% reported it was easy to connect to the live baby fair presentation using a video conferencing platform. For baby fairs with a length of 1 h and 15 min, 64% of respondents felt that the time was perfect, while 36% would attend a longer presentation where more information was presented. For baby fairs with a length of 2 h, 100% of respondents thought the length of time was perfect. No respondent reported that the length of the baby fair was too long. Most respondents, 91%, found the baby fair resource packet helpful, while 9% said it was not.

## 4. Discussion

As a result of the COVID-19 pandemic, the Michigan NBS Program developed a virtual education initiative that could be useful for all NBS programs as a tool to educate parents prenatally about NBS despite limited resources. The virtual baby fair allowed the Michigan NBS Program to distribute important information about NBS to parents and provided parents access to a plethora of health and safety information and a platform to ask questions. By providing a written resource, a live presentation, and a recording of the presentation on YouTube, parents had the opportunity to learn at their own pace using their preferred platform.

A benefit to the virtual format was that expecting parents from a wide geographic range attended. More counties were represented than Michigan’s NBS Program had been able to reach with previous in-person baby fairs. From 2016–2020, Michigan NBS staff attended 16 in-person baby fairs in 8 counties (Figure 1a). The virtual baby fair attracted registrants from 44 counties with only 7 virtual fairs (Figure 1b). Additionally, of these 44 counties with at least one registrant, 35 contain MUAs. Thus, virtual baby fairs could potentially result in making access to NBS education more equitable. However, only county of residence was collected from participants, so we are not able to determine if attendees resided in MUAs within those counties.

The baby fair was planned, organized, and conducted with little financial support. The NBS program covered the cost of postage for mailing the requested printed resources (under $100). In-person baby fairs typically cost about $1000 on average when including staff time and vendor fees. One staff member from the NBS program staff led coordination of the event, and one additional staff member assisted with logistics. The time commitment was most significant for the coordinator of the event, with most of the work being completed in the month prior to the event. This time was able to be incorporated into the time typically allotted for in-person education events. Additional staff time required of other participating program staff was minimal, with most of the work for them involving the assembly of the program page in the resource packet, attending the live presentation, and promotion of the event. The time commitment for all staff decreased with each event as the framework created for the inaugural event was used for subsequent baby fairs. 

The Michigan NBS Program experienced a few limitations with the baby fairs. To date, the Virtual Baby Fair has only been presented in English, which creates barriers for people who do not speak English. For all seven fairs, a low percent of registrants (23%) attended the live presentations. Additionally, the low response rate on the post-event surveys (6%) could limit the planning committee’s ability to implement changes that would make the event experience better, as the responses only represent a small percent of total registrants and attendees. Finally, the participating programs were limited to a five-minute presentation, which is a small amount of time to fit all the information that might be useful for parents to know about their program.

Despite these limitations, the planning committee learned ways to improve the event with each fair. After the first round of baby fairs, the planning committee increased the length of the event to two hours and added more programs. Feedback from post-event surveys indicated that families thought this amount of time was appropriate. The committee also identified additional areas for improvement for future fairs. For the next round of virtual fairs, the committee will test the use of an event platform for registration that sends calendar reminders and instant emails to registrants following registration to confirm their registration. This could help increase the percent of registrants that attend the live presentation. Making the baby fair and resource packet and presentation available in more languages would make the baby fair accessible to a wider audience. This could potentially be accomplished by reaching out to local translation support organizations that might be willing to volunteer their time to help make it accessible to those who could benefit from the presentation in a different language. Additionally, to increase the attendance, the planning committee could consider applying for grant funding to support advertisement of the event on a larger scale and promoting the event to more prenatal care providers to distribute to parents.

A future step for the Michigan NBS Program is to assess the impact of the virtual baby fair on parent knowledge of NBS. This could be done by surveying registrants with a pre- and post-event survey. The survey could allow the Michigan NBS Program to determine whether the baby fair is an effective way to deliver information to improve parental understanding of NBS. Questions could be included on the survey to help understand what programs were most helpful to expecting parents and encouraged their attendance.

## 5. Conclusions

The virtual baby fair allowed the NBS program to educate more parents across Michigan than attending in-person events had in the past, at almost no additional cost to the program. This is especially important for smaller programs with limited staff and budget for education. The virtual format also allowed for expecting parents from a wide geographic to range to attend. The NBS program could have led this initiative alone and only educated parents exclusively about NBS. By taking a team approach to the baby fair and partnering with other programs, the event was promoted to a larger audience and potentially attracted more parents that might not otherwise seek out NBS education. Moving forward, the Michigan NBS Program plans to host virtual baby fairs twice a year to give every new and expecting parent the chance to receive information they need to prepare for their baby’s newborn screen.

## Figures and Tables

**Figure 1 IJNS-08-00034-f001:**
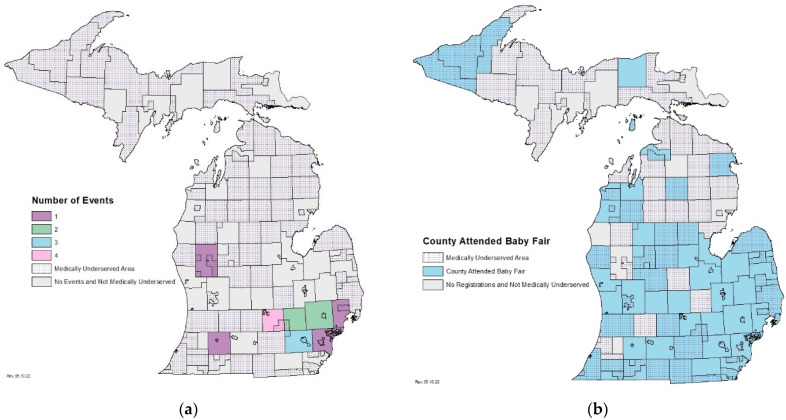
Geographic reach of the NBS program educational efforts: (**a**) Counties where the Michigan NBS Program was in attendance as a vendor at an in-person educational event. (**b**) Counties in attendance at the Michigan Virtual Baby Fairs from 2020–2021.

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
