# Peer review of "Use of Online Newborn Screening Educational Resources for the Education of Expectant Parents: An Improvement in Equity"

_2409-515X, 2022, doi:10.3390/ijns8020034_

Round 1
Reviewer 1 Report
This is an interesting and well written paper. No further suggestions.
Reviewer 2 Report
Suggestions regarding the title: While the pandemic may have been the stimulus for initiating the author’s on-line outreach, I’m not sure that it needs to be in the title. Similarly, the “further together” and partnership with other public health programs are less the key point of the author’s observations than that the offering of on-line educational materials to expectant parents may provide a wider exposure than prior in-person fairs. A better focused title might be something along the lines of “Use of on-line newborn screening educational resources for the education of expectant parents: an improvement in equity”.
Making sure that parents are adequately educated regarding the reasons for and the process of newborn screening for their new baby is a laudable goal. Each state department of public health has a unique newborn screening program and approaches education of parents in their own way based on the available resources, the population served and the creativity of the staff. Many states simply distribute brochures, perhaps because of registration for obstetric care or when admitted for delivery, and most programs maintain a website that provides information for self-education. Unfortunately, many factors must align for this information to be effectively transmitted; patients not in prenatal care may not be accessible prior to delivery, parents given brochures may not read them (particularly when included in a sea of other prenatal information), and even if parents are exposed to the information, they may not fully understand the implications of public health mandates, consents, pilots, biobanks, the underlying reason for why NBS is done or how outcomes may be improved and/or lives saved through NBS.
It is reported that the department of public health staff had previously offered information at in-person baby fairs, but no information is supplied on the success of uptake (e.g. number of registrants or attendees) to those events, thus not allowing for an in-person vs. virtual comparison.
There was also likely pre-existing distribution of information to expectant parents through brochures or website visits. The background in the manuscript includes no description of the preexisting parental education program, with exception of the baby fairs. Presumably, the baby fairs were just one prong of a multi-pronged approach to getting information to parents on why newborn screening is important, what is screened for, and what happens in the pre-analytic through post-analytic phases of the process. This is not quantitated or estimated as part of a full disclosure of all modes of providing educational materials on NBS, which would give a more complete educational picture. If a pie were to represent educating all expectant parents on NBS, what proportion would have been attributable to routinely distributed brochures, DPH website hits and in-person baby fairs prior to and after institution of the virtual fairs?
Placing their “fair booth” on-line, inviting parents to observe a presentation (and allowing them to ask questions) is an excellent use of technology for education. Unfortunately, the number of parents reached by the described programs is tiny in comparison to the birthrate in Michigan. The authors should present their uptake (e.g. registriation, documented access to on-line materials, questions), both before and after institution of the on-line fairs. If there are tens of thousands of births, but only a fraction of a percent of parents actually participating with the described on-line programs, on-line presentation or viewed a recording of that presentation, there is still a massive gap remaining that needs to be filled, as all parents should be made aware of NBS and what it could mean to them.
Michigan has established a superb biobank based on retained DBS. This adds a unique requirement for education for consenting – a separate issue from the purpose of NBS itself. The manuscript does not try to tease apart the generalizable issues of educating parents on basic NBS and the Michigan specific need to inform parents about the biobank for their consent. There is no discussion about the broader issue of the ultimate destination of the dried blood spot after testing, which has been controversial and has state specific issues. Most states have no biobank, so this is not an issue. Retention of DBS for QI or research is dealt with differently by each state. If there is no retention, the issue is not needed. If there is no biobank, that topic does not need to be addressed. If a state retains DBS for both programmatic QI and allowance of some IRB approved research, this might require a state specific explanation, even without the need for a documented signed consent as a next step. Some states may just include a statement in the brochure distributed at birth that states the policy for DBS handing, yet allows the parents to opt-out if they so request.
There is no assessment of the impact of in-person vs. virtual exposure (e.g knowledge evaluation through the post-survey). Because there was none performed in the pre-phase, there would be no comparators, but at least there could have been an estimate of the efficacy of the virtual intervention to assess whether the presentation was effective.
The background is also weak in justifying, based on existing literature, why it is important for parents to be provided the information content presented in the virtual fair, or whether providing such information impacts uptake (either of mandated NBS or biobank enrollment) or understanding.
How might the developed resources be utilized for education of providers (both pediatricians and obstetricians)? There is some background provided regarding recommendations for OB providers, but no estimate for Michigan about what is done currently, or what the impact of the virtual fairs has been on fulfilment of the described ACOG recommendations.
When the authors state that previous Michigan research showed that prenatal education on the Biotrust resulted in more completion of consent forms, does that just mean return of the form with some response (Yes or No) or that parents were more likely to say Yes after the intervention (p.2 line 75)
With the in-person fair, was there any collection of zipcode or county data on attendants for comparison with the virtual attendees?
Why didn’t the authors post the recording of the virtual fair to the general public or distribute that in a link via their standard prenatal brochure? The manuscript implies that the recoding was only made available to registrants and attendees. Was there a reason for restricting distribution of the information, and perhaps including a contact address for questions that would reach a wider audience?
How do the authors quantitate the benefit of linking the NBS portion to the other “team” components? Would the virtual fair have been just as impactful as a standalone offer? Is the audience the same people who are interested in lead poisoning – one would think this would be a later interest, after the fact of doing NBS. Might the inclusion of multiple unrelated programs be a distraction from parents who wanted to focus on NBS information, or did the other programs have more of a primary interest to parents and that brought exposure to NBS that would not have happened if the parents were not looking for that unrelated information? Did the survey ask what brought the parents to the virtual fair as their primary interest?
The topics of the other fair presenters should be included with the description of the methods instead of the result section, rather than just “programs within the MDHSS with a similar target audience” or “other areas of MDHHS”. The methods section is where the reader wants to understand all the components of the project, and using those general terms makes the reader ask questions about who else was involved that are distracting. The answers to the reader’s question are answered in the results section, but by then the reader has wasted time trying to answer the question by looking around in the manuscript.
Were there different advertising methods used for the in-person fairs, and if so, what were they?
Did the survey ask for any assessment of the written materials that were distributed for the virtual fair?
Is there any estimate of the actual amount of time spent on the in-person vs. the virtual fairs?
It appears that “registrants” and “attendees” are separate cohorts. Does p.6 line 237 really mean “attendee” or could it be registrant? How do the numbers differ in MUA distribution for registrants and attendees?
Why do the authors think the response rate to the surveys is so low?
Given there have been repeat virtual fairs, have the authors utilized any PDSA cycle methodology to evaluate improvement over repeat presentations based on lessons learned that have resulted in interventions for modifications?
Was there an additional cost for mailings for the virtual fairs (postage, printing) vs. the in-person fairs?
Reviewer 3 Report
The authors describe an approach to transmit information and education about the NBS program to parents, during the COVID pandemic. However, the article would be better readable and understandable, if the authors would provide a link to a recorded baby fair.
The 2 links in supplemental material could be deleted, since they do not provide any substantial information.
Round 2
Reviewer 2 Report
Comments on the prior version have been adequately addressed.